# Low Dietary Diversity and Low Haemoglobin Status in Ghanaian Female Boarding and Day Senior High School Students: A Cross-Sectional Study

**DOI:** 10.3390/medicina60071045

**Published:** 2024-06-26

**Authors:** Joyce Asare, Jia Jiet Lim, Isaac Amoah

**Affiliations:** 1Department of Biochemistry and Biotechnology, Kwame Nkrumah University of Science and Technology, Kumasi 00233, Ghana; jasare25@idl.knust.edu.gh; 2Human Nutrition Unit, School of Biological Sciences, University of Auckland, Auckland 1024, New Zealand; jlim287@aucklanduni.ac.nz

**Keywords:** adolescents, anaemia, dietary diversity score, Ghana, haemoglobin, students

## Abstract

*Background and Objectives*: Anaemia is one of the most common forms of malnutrition globally, with most anaemia cases related to micronutrient deficiency. Diets with higher dietary diversity scores (DDS) are usually nutritionally diverse and could positively impact micronutrient status, including haemoglobin (Hb) concentration improvement. The study aimed to determine DDS and its association with the Hb concentration of Boarding and Day adolescent Senior High School students in Ghana. *Materials and Methods*: A semi-structured and three 24 h dietary recalls were used to obtain the participants’ demographic and diet intake data, respectively. Hb concentration was assessed using a validated portable haemoglobinometer. DDS was evaluated using the Minimum Dietary Diversity for Women (MDD-W) approach. *Results*: A significant difference in the DDS between Boarding and Day students existed. Only 22% of the Boarding students had adequate dietary diversity, whereas 64% of the Day students had adequate dietary diversity. A significantly smaller proportion of the Boarding students consumed nuts and seeds, dairy, flesh foods, eggs, vitamin A-rich vegetables and fruits, other vegetables, and other fruits compared to Day students (*p* < 0.05, all). No significant difference (*p* = 0.925) in mean (±SD) Hb concentrations between Boarding (11.9 ± 1.1 g/dL) and Day (11.9 ± 1.1 g/dL) students was found. Additionally, no significant correlation between mean DDS and Hb concentration (*p* = 0.997) was recorded. Using Hb < 12 g/dL as the determination of anaemia, 55.1% Boarding and 57.8% Day students had anaemia. *Conclusions*: Low dietary diversity in Boarding students highlighted inadequate nutrition provided by school meals. Strategies to increase meal diversity should be prioritised by stakeholders in Ghana’s educational sector.

## 1. Introduction

Adolescence is a critical time for growth and development, leading into adulthood. Good nutrition during this period is vital for ensuring healthy development [1,2]. Despite the critical nature of the adolescence phase, many adolescents in developing countries, including Ghana, are facing challenges in consuming a diverse and nutritious diet [2].

Children and adolescents who commonly consume insufficient animal-based foods, fruits and vegetables, but excessive convenient, energy-dense nutrient-depleted foods [3] may suffer from hidden hunger characterised by micronutrient deficiency [4,5]. Occasionally, they may be apparently “well-fed”, hence overweight, but also suffer from micronutrient deficiency [6]. Consequently, the consumption of a high-quality diet composed of foods from diverse food groups to ensure sufficient intake of various micronutrients is greatly encouraged [7,8].

In the year 2017, a national survey, which took the form of a cross-sectional study was conducted using non-pregnant adolescent women aged 15–49 years in Ghana to determine micronutrient deficiencies [9]. The authors reported that the prevalence of anaemia, vitamin A deficiency, folate deficiency and vitamin B12 deficiency stood at 21.7%, 1.5%, 53.8%, and 6.9%, respectively [9].

Anaemia remains one of the most common forms of malnutrition globally. Using the World Health Organisation’s (WHO) threshold for diagnosing anaemia based on haemoglobin (Hb) concentration, nearly one-third of female adolescents aged 15–19 years are affected by anaemia globally, and the prevalence remains high throughout childbearing age [10]. Based on the same threshold, the recent Ghana Demographic and Health Survey report revealed that 41% of non-pregnant Ghanaian women aged 15–49 years were anaemic [11], higher than the global average. Another recent cross-sectional study in Ghana involving an analysis of data from students comprising 2948 adolescent females and 609 males aged 10–19 years showed an anaemia prevalence of 24 and 13%, respectively [12]. Whilst there are many causes of anaemia, more than half of the anaemia cases globally, including in African countries, are attributable to dietary iron deficiency [5].

Adverse effects associated with anaemia include attention deficit, poor cognitive ability and overall poor academic performance [13,14]. Consequently, an understanding of the dietary habits among female school-going adolescents will pave the way for identifying future school-based dietary interventions to decrease the prevalence of anaemia.

School is the perfect setting for dietary intervention as it delivers food to all school-going children and adolescents. For example, the school feeding program has been very successful in providing food security and decreasing the prevalence of stunting in low- and middle-income countries [15]. In Ghana, the educational structure preceding the tertiary level is commonly referred to as Senior High School, where students are presented with two residential options, either Boarding or Day. The Boarding students reside within the school premises, where they receive their meals from the school’s dining facility, with the option to supplement their diets with purchased foods. In contrast, the Day students attend classes in the school and return to their homes daily, consuming meals primarily prepared at home or outside the school premises. Consequently, the nutrition adequacy of Boarding students is largely determined by the foods provided by the school. Nevertheless, the adequacy of dietary diversity of Boarding students in Ghanaian public schools is questionable as the food provision is often limited by the financial constraints of the funding body [16].

In China, it has been reported that dietary staples for Boarding students are predominantly rich in carbohydrates and inferior in protein-rich food. This may potentially predispose the students to anaemia as proteins remain an integral component of haematinic diets especially when taken with vitamin C [17]. In Ghana, it has been established that poor dietary diversity score is not an important predictor of anaemia among children aged 6–59 months in Ghana [18]. What is however unknown is the association between dietary diversity score and anaemia in adolescents Boarding and Day students.

In this study, we aim to compare the dietary diversity of Boarding and Day students in a metropolitan public female Senior High School, and the association between dietary diversity and Hb concentration. Whilst there are existing data on the association between dietary diversity and health status, this study uniquely introduces the residential status of students as a contributing factor to dietary diversity and health status. Despite the students accessing the same education environment, the food environment between Boarding and Day students may be different. This study will offer a novel understanding of the importance of school meal provision on the dietary diversity of Boarding students. Most importantly, we hypothesised that students with higher dietary diversity will have higher Hb concentration.

## 2. Materials and Methods

### 2.1. Study Design

This is a cross-sectional study conducted at the Tema Senior High School. Data were collected between February and April 2023.

### 2.2. Study Site and Sample Size Calculation

The study was conducted at Tema Senior High School in the Tema West Municipality. The school includes both Day and Boarding options [19]. It is the oldest school in the Tema Metropolis [19] and has the largest adolescent student population, making it a representative senior high school in the municipality.

The sample size was calculated using the Taro Yamane’s;
$$n=\frac{N}{{1+(Ne}^{2})}$$
where n is the sample size required for the study; N is the total population size and e is the desired level of precision (5% = 0.05) at 95% confidence interval. The N refers to the total number of students aged ≥ 15 years and above which is 576. This age group was targeted because the MDDS-W requires that women aged ≥ 15 to 49 years be used. Additionally, in the recent Ghana Demographic and Health survey reports, the age group of women selected for anaemia testing was from 15 to 49 years [11].
$$\begin{array}{l} n=\frac{576}{{1+(576\;\left(0.05\right)}^{2})}\\ n=236 \end{array}$$

Thus, the minimum number of participants required for the study is 236.

### 2.3. Participants

Participants included in this study were adolescent females aged between 15 and 19 years. The Boarding and Day students were recruited at 1:1 ratio using convenience sampling method. Students who were taking iron supplements and had sickle cell anaemia based on self-report were excluded. All students and their parents/guardians provided written informed consent.

### 2.4. Data Collection

A semi-structured questionnaire was used to collect participant information on sociodemographic characteristics. A trained dietitian collected data from each student through a combination of self-administered questionnaires and face-to-face interviews.

### 2.5. Anthropometric and Biochemical Assessment

Body weight was measured to the closest 0.1 kg with the students standing on a body composition monitor (Omron, BF511, Kyoto, Japan) barefooted and in light clothes. Height was measured to the closest 0.5 cm with the students standing in an erect position against a stadiometer’s vertical scale (Seca, Hamburg, Germany). Body mass index (BMI) of students was recorded to the nearest one decimal place. Stunting and BMI status (severe thinness, thinness, normal, overweight, and obese) were categorised using the WHO’s height-for-age and BMI-for-age charts, respectively [20].

### 2.6. Haemoglobin Measurements

The Hb concentration of the students was determined using a digital haemoglobinometer (GLC-HGB-21), validated against the gold standard Sysmex (XN-330) Hb auto-analyser in the laboratory. The validation involved a comparative analysis of venous blood Hb concentrations in thirty participants. A strong association (*r*^2^ = 0.893) between the digital haemoglobinometer measures and the laboratory-based measures was established (Appendix A). This confirmed the reliability of using the digital haemoglobinometer for large population-based studies to measure Hb concentrations in the field. In the present study, venous blood samples of students were collected by a phlebotomist at the school premises into EDTA vacutainer. The Hb concentrations of the anticoagulated blood were determined using the validated digital haemoglobinometer in measuring time of <10 s. According to the WHO, anaemia is defined as having Hb ≤ 12.0 g/dL [21].

### 2.7. Dietary Diversity Score Determination

Three 24 h dietary recalls were conducted by a trained dietitian for each student. The use of the three 24 h dietary recalls was to provide a broader picture of the dietary intake of the students as against the use of one 24 h recall. These recalls included two weekdays and one weekend day, to capture a comprehensive view of the students’ dietary habits during the weekdays, including any variations that might occur on weekends. Students were guided through a structured dietary recall table, during which they were asked to recall all foods and beverages consumed over the selected days. Students were asked to estimate the portion sizes of the foods consumed. To facilitate detailed recall, the trained dietitian used probing techniques, asking about specific meals, snacks, and beverages consumed.

The dietary data collected from the recalls were used to determine the dietary diversity score following the Minimum Dietary Diversity for Women (MDD-W) approach, validated by the FAO and FHI [22]. The MDD-W was designed for females in reproductive stage aged 15–49 years. The MDD-W framework categorises foods into 10 distinct groups (Appendix A). A score of 1 was assigned for each food group when the student reported consuming at least 15 g of food within the food group over a 24 h recall period, allowing for a possible score range of 0 to 10. A score of 10 indicates a consumption of all food groups, reflecting maximal dietary diversity. For each student, the dietary diversity score (DDS) was calculated for each 24 h dietary recall period. Subsequently, a mean DDS over the three 24 h dietary recall periods was calculated, whereby the student was considered as achieving adequate dietary diversity when the mean DDS > 5.

### 2.8. Statistical Analysis

Data analysis was carried out using International Business Corporation SPSS Statistics (version 20.0, New York, NY, USA). Continuous data were checked for normality using the Kolmogorov–Smirnov test. Participant characteristics were reported as mean ± standard deviation (SD) using descriptive statistics. Differences in dietary diversity score and Hb outcomes between Boarding and Day students were compared using independent sample *t*-test. Pearson’s correlation was used to test the correlation between Hb and mean dietary diversity score.

We employed Generalised Estimating Equations (GEE) to examine the differences in the consumption of each food group between Boarding and Day students over the three diet recall days. GEE accommodates the binary nature of outcome variable (yes/no) and accounts for the correlated data arising from repeated data collections on the same participants. Each individual participant was included as the subject variable, time of data collection (Weekday 1, Weekday 2, and Weekend) was included as the repeated variable, the binary response of food consumption was included as the dependent variable, and the boarding status (Border/Day) was included as the predictor variable. The significance of predictor variable was tested using Wald Chi-Square statistic, effect size was presented as odds ratio (OR) with 95% confidence interval (CI). To explore whether the consumption of each food group was different between Hb status, the GEE was repeated using the Hb status (adequate Hb/Low Hb) as the predictor variable. A significance level of α = 0.05 was applied for all statistical tests.

We utilise ChatGPT-4 to visualise the data on the consumption of each food group. Raw data were uploaded to ChatGPT-4. ChatGPT-4 was instructed to generate radar charts whereby the angular axis represents food groups and radial axis represents the percentage of population reported to consume the food group.

## 3. Results

A total of 118 Boarding and 118 Day students were included in this cross-sectional study. Despite the narrow age range, the average age of the Boarding students (16.9 ± 1.4 years) was significantly higher than that of the Day students (16.5 ± 1.1 years) (Table 1). In terms of ethnicity, out of the 75 native ethnic groups in Ghana, the participants identified themselves with seventeen different groups (Appendix A).

The Boarding students also recorded significantly higher BMI (22.8 ± 3.8 years) than the Day students (21.7 ± 4.1 years) (Table 1). In this cohort, three Boarding students were classified as stunting, whereas one Boarding student was classified as severe stunting. In contrast, one Day student was classified as severe stunting.

The mean DDS was significantly lower for the Boarding students compared to the Day students (Table 2).

Adequate dietary diversity is defined as mean DDS >5. Using this definition, only 26 out of 118 participants (22%) in the Boarding students had adequate dietary diversity, whereas 75 out of 118 participants (64%) in the Day students had adequate dietary diversity.

The Boarding students were less likely to consume nuts and seeds (OR: 0.36, 95%CI [0.20–0.66], *p <* 0.001), dairy (OR: 0.45, 95%CI [0.29–0.70], *p* < 0.001), flesh foods (OR: 0.38, 95%CI [0.28–0.50], *p* < 0.001), eggs (OR: 0.59, 95%CI [0.41–0.85], *p* = 0.005), vitamin A-rich vegetables and fruits (OR: 0.49, 95%CI [0.35–0.69], *p* < 0.001), other vegetables (OR: 0.57, 95%CI [0.38–0.85], *p* = 0.006), and other fruits (OR: 0.21, 95%CI [0.10, 0.43], *p* < 0.001) compared to the Day students (Figure 1). A more concerning outcome was that the consumption of nuts and seeds, dairy, eggs, dark green leafy vegetables, and other fruits, was consistently low across the three food recall days in this cohort.

Despite mean DDS being significantly lower in the Boarding students, there were no significant differences in mean (±SD) Hb concentrations between Boarding (11.9 ± 1.1 g/dL) and Day (11.9 ± 1.1 g/dL) students (*p* = 0.925) (Figure 2).

Using Hb < 12 g/dL as the determination of anaemia, we found that 55.1% of Boarding and 57.8% of Day students had anaemia. There was no significant correlation between Hb concentration and mean DDS (*p* = 0.997) (Figure 3).

The result remained the same when controlled for Boarding status using partial correlation analysis (*p* = 0.971). There was also no significant correlation between mean DDS and Hb concentration (*p* = 0.997) (Figure 3). Although about half of the students in this cohort had low Hb (anaemia), our exploratory analysis revealed there were no significant differences in the consumption of each food group between students with and without anaemia (*p* > 0.05, all) (Figure 4).

## 4. Discussion

Our study findings reveal a notable disparity in dietary diversity between Boarding and Day students, with the Boarding students exhibiting significantly lower dietary diversity. This is evidenced by the lack of consumption of certain food groups such as nuts and seeds, dairy products, flesh foods, eggs, and vitamin A-rich vegetables and fruits, as well as other vegetables and other fruits among Boarding students compared to Day students. Despite these differences in dietary intake, both groups displayed similar mean Hb concentrations. Nevertheless, the prevalence of anaemia was considerably high in both groups, with 55.1% in Boarding students and 57.8% in Day students. This outcome did not align with our initial hypothesis, which posited that higher dietary diversity would positively correlate with increased Hb concentrations. Moreover, the contrast in dietary patterns was more pronounced when comparing Boarding and Day students directly, rather than when comparing students based on the presence or absence of anaemia.

The most important outcome of this study is the revelation that Boarding students in this metropolitan Ghanaian public high school had lower dietary diversity than Day students. Based on the dietary recalls of Boarding students, it likely reflected that school meals were lacking protein-rich foods such as nuts and seeds, dairy, flesh foods, and eggs, as well as micronutrient-rich plant foods such as vitamin A-rich vegetables and fruits, other vegetables, and other fruits. This observation is consistent with a previous report in China that school meals for Boarding students may be monotonous and rich in carbohydrate staples [17]. Therefore, it becomes a voluntary choice for Boarding students to purchase additional foods to improve their dietary diversity. Nevertheless, the consumption of nuts and seeds, dairy, eggs, dark green leafy vegetables, and other fruits was generally low in this cohort, which is also consistent with that reported by other Sub-Saharan African school-going adolescents [23].

We also advise caution when interpreting the outcomes comparing Boarding and Day students. The diversity of school meals is likely to be influenced by institutional funding, which varies from school to school. Furthermore, the dietary diversity of Day students is likely to be influenced by the socioeconomic factors of the student’s family. In this cohort, the higher dietary diversity of Day students, in combination with the low prevalence of stunting and underweight, suggested that they were unlikely to face food insecurity issues. Also, seasonal variations in diet may affect the dietary diversity of the students. Another Ghanaian report revealed no significant differences in macronutrient and micronutrient intake between Boarding and Day students in three private elementary schools in Accra Metropolis, but it was limited by a small number of Boarding students (n = 30) relative to Day students (n = 94) [16].

Concerning anaemia, it remains an important public health challenge in Ghana, as well as the wider Sub-Saharan Africa. In the present study, more than half of the students had anaemia defined as Hb concentration ≤ 12.0 g/dL. Despite the lower dietary diversity in the Boarding students, we did not detect a significant difference in mean Hb concentration between Boarding and Day students. Similarly, we did not detect a significant difference in dietary diversity between students with and without anaemia. This could be explained by that low Hb concentration is usually developed over a prolonged period of undernutrition and hence may not be simply captured by dietary recalls at a single point in time. Long-term dietary diversity may be a better indicator of Hb concentration. Furthermore, a quantitative dietary analysis may have captured more information compared to using dietary diversity analysis, but we were limited by the lack of a reliable food composition database for Ghana at the time of this study [24]. Although known sickle-cell anaemia is an exclusion criterion for participating in this study, it is possible that other undiagnosed parasitic infections in the students may influence the Hb concentration.

In contrast to our hypothesis, cross-sectional evaluation of dietary diversity is not an indicator of the anaemia status at present. The 2014 Ghana Demographic and Health survey data showed that dietary diversity was positively associated with mean Hb concentrations among children 6–23 months, yet it was negatively associated with Hb concentrations among children 48–59 months [18]. In terms of anaemia prevalence, a weak positive association between minimum dietary diversity and anaemia prevalence was observed only in children aged 6–23 months [18]. In a cross-sectional study conducted in the northern part of Ghana involving 400 pregnant women, no significant association between dietary diversity and Hb of the pregnant women [25]. In both studies mentioned [18,23], <50% of participants had adequate dietary diversity, but >50% of participants had anaemia based on Hb concentration. These prevalences are similar to our current study. Despite the lack of consistent cross-sectional association between dietary diversity and Hb concentration, low dietary diversity may put them at risk of future anaemia. Therefore, a longitudinal observation may have been more informative.

The most common dietary factors for preventing and managing anaemia include flesh foods, fruits and vegetables consumption [26]. Red meat contains heme iron, which is highly bioavailable, whereas absorption of non-heme iron from plant foods can be enhanced by the co-ingestion of flesh meat and vitamin C-rich fruits and vegetables [27]. In Ghana, muscle meat which contains bioavailable forms of haem iron is relatively expensive. In contrast, plant-based alternative sources, especially from indigenous green leafy vegetables tend to be the common source of non-heme iron for the older population. However, in recent times, there has been a general decline in indigenous leafy vegetable intake particularly amongst adolescent Ghanaians. This could have contributed to the high anaemia prevalence. In addition to the low intake of leafy vegetables, factors including poor cooking practices for indigenous vegetables, overcooking of indigenous vegetables and poor intake of vitamin C-rich fruit decrease the bioavailability of iron for absorption. In short, the dietary diversity of both Boarding and Day students was not favourable for preventing iron-deficiency anaemia in the long term. Although this MDD-W approach did not evaluate coffee and tea intake, which contains inhibitors for iron absorption, our 24 h dietary recalls showed that they were not frequent consumers of tea and coffee. This study design could be improved by concurrent measurement of serum iron and ferritin, which would be an indicator of iron-deficiency anaemia. Despite iron-deficiency anaemia being the most common form of anaemia globally, being female, menstruation and parasitic infection were also directly associated with the apparently high anaemia prevalence amongst adolescents in sub-Saharan Africa [28].

In addressing the high prevalence of anaemia among both boarding and day students, a multifaceted approach is necessary. This includes the introduction of educational programs focused on the importance of consuming iron-rich foods and those that facilitate iron absorption, such as flesh foods, citrus fruits and dark green leafy vegetables. Furthermore, the Girls’ Iron–Folic Acid Tablet Supplementation (GIFTS) Program has also been in place since October 2017 to decrease the prevalence of anaemia among Ghanaian school-going girls and female adolescents. GIFTS combines nutrition education with weekly provision of iron–folic acid tablet supplementation over 30–36 weeks in each academic year to eligible female students between 10 and 19 years, successfully reaching approximately 400,000 female students. A prospective analysis of the GIFTS program revealed that 51% of female students who had anaemia at the start of the academic year no longer had anaemia at the end of the academic year following the intervention, which was a great achievement [29]. Yet, we have realised that GIFTS is currently unsustainable as there is a shortage of iron–folic acid tablets in Ghana currently. Notably, students in the current study are not currently benefiting from GIFTS.

### Limitation

This study is limited by being conducted in a single metropolitan public high school in Ghana. The findings may not be generalisable to other regions or to private schools, where dietary patterns, socioeconomic factors, and school funding might differ. The choice of one Senior High School for this study limits the implications of the study to just the school and may not be a full representation of the broader picture in the Greater Accra region of Ghana. In terms of sociodemographic variables, only ethnicity was obtained from the students. Nevertheless, other socio-demographic variables such as household size, occupation of parents, education level of parents, etc., are not expected to influence the dietary diversity of the boarding students. Hence, these variables were not recorded.

The cross-sectional nature of the study also limits the ability to establish causal relationships between dietary diversity and Hb status, and we have suggested that longitudinal studies would be more effective in understanding the impact of dietary patterns over time. Whilst the convenience sampling method is a common recruitment method, it is not as robust as the random sampling method used in large-scale epidemiology studies. Finally, there is no perfect method to evaluate dietary intake. In this study, the reliance on self-reported dietary intake can introduce recall bias, particularly in adolescent populations. The accuracy of the dietary recall data is dependent on the participants’ memory and honesty.

## 5. Conclusions

In conclusion, the study has illuminated critical insights into the dietary habits and nutritional status of Boarding and Day students in a metropolitan Ghanaian public high school, underscoring a significant disparity in dietary diversity between the two groups. Despite the lower dietary diversity observed among Boarding students, there was no significant difference in Hb concentrations between Boarding and Day students, challenging the initial hypothesis that higher dietary diversity would correlate with increased Hb concentration. Nevertheless, it is equally important to note that the dietary diversity of both Boarding and Day students remains sub-optimal and that the anaemia prevalence of both Boarding and Day students is equally high. Despite national nutritional strategies such as nutrition-friendly school programs and GIFTS in place, we did not find evidence of these programs benefiting the students included in this study. Since Boarding students received most of their meals from the school premise, their low dietary diversity likely reflects the inadequate nutrition content of school meals.

### Recommendations

There is a need to enhance the nutritional profile of meals served to senior high school students by incorporating foods rich in bioavailable iron. Additionally, school food producers should be trained on strategies for cooking plant-based legumes, which can enhance the digestibility of proteins. Promoting the increased consumption of dark green indigenous vegetables should also be a priority, as these vegetables provide relatively cheaper sources of iron. In recent times, fish farming especially involving catfish and tilapia rearing is on the rise in Ghana. Schools can be encouraged to adopt that as part of their science curriculum which could provide a pathway for fish meal powders which could be promoted for its use in cooking recipes. Lastly, adding vitamin C-rich fruits to school-based meals should be encouraged to enhance the absorption of dietary iron.

## Figures and Tables

**Figure 1 medicina-60-01045-f001:**
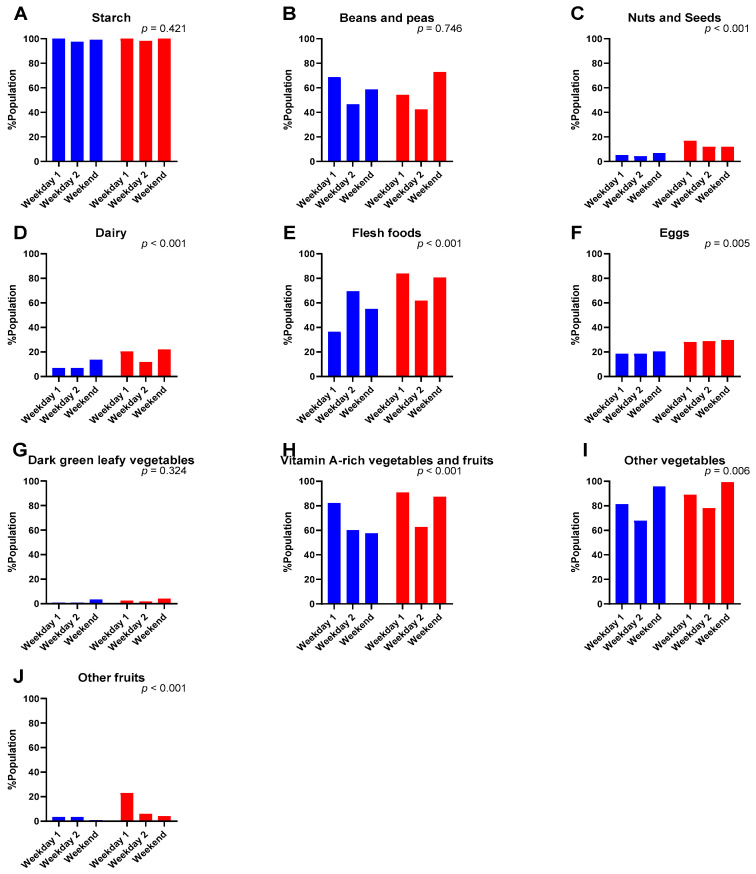
Percentage of population consuming each food group at each food recall day (Weekday 1, Weekday 2, and Weekend). Blue represents Boarding students, red represents Day students. Differences in the consumption of each food group between the Boarding and Day students were compared using Generalised Estimating Equations.

**Figure 2 medicina-60-01045-f002:**
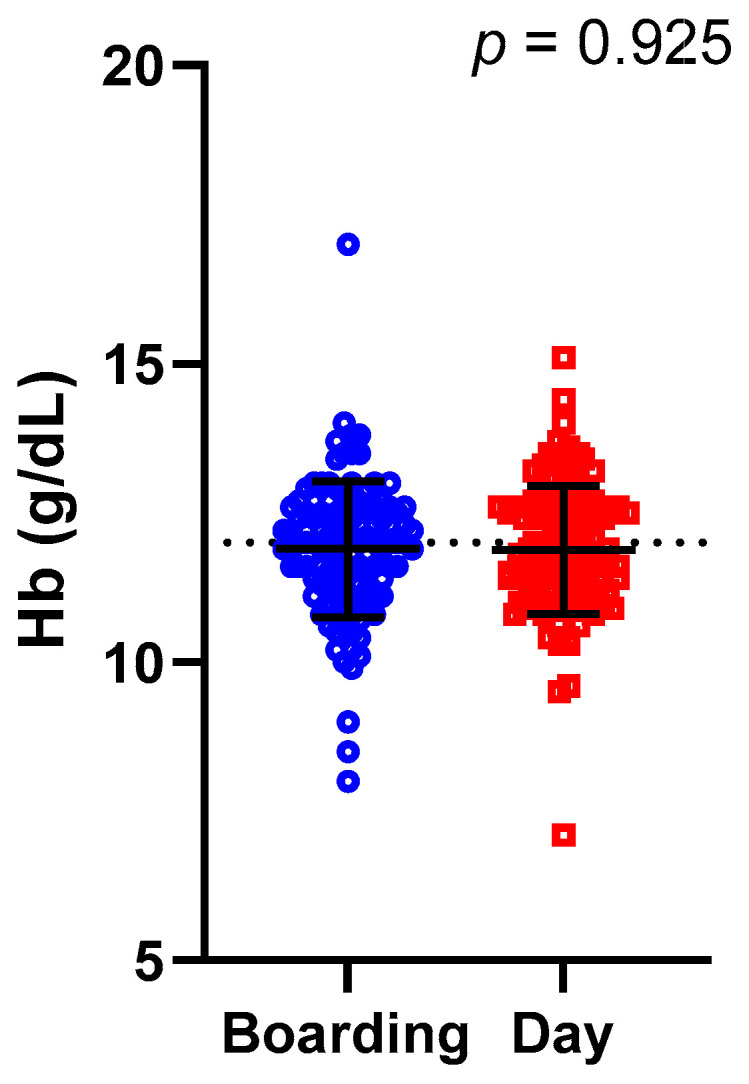
Hb concentration grouped by Boarding and Day. The difference in Hb between Boarding and Day students is compared using independent sample *t*-test. Individual data is presented, alongside mean ± SD as error bars. Horizontal dot represents adequate Hb concentration (12 g/dL).

**Figure 3 medicina-60-01045-f003:**
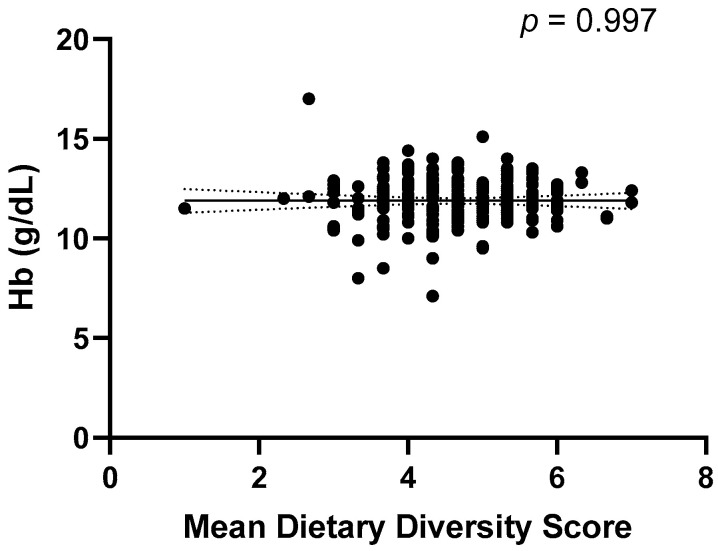
A scatterplot of Hb against mean dietary diversity score with best-fit line and 95% confidence interval. The correlation between mean dietary diversity score and Hb was tested using Pearson’s correlation.

**Figure 4 medicina-60-01045-f004:**
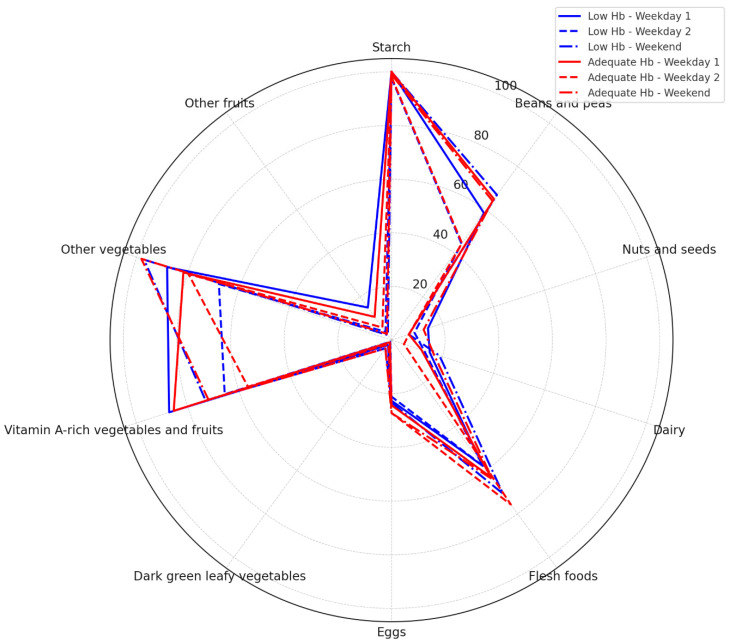
Radar percentage of population consuming each food group at each 24 h dietary recall day (Weekday 1, Weekday 2, and Weekend), grouped by Hb status.

**Table 1 medicina-60-01045-t001:** General characteristics of participants.

Parameters	Boarding Students (n = 118)	Day Students (n = 118)	*p*-Value
Age (yr)	16.9 ± 1.4	16.5 ± 1.1	0.011
Weight (kg)	58.5 ± 10.3	55.8 ± 11.5	0.059
Height (m)	1.602 ± 0.056	1.603 ± 0.055	0.801
Stunting (n)	3	0	
Severe stunting (n)	1	1	
BMI (kg/m^2^)	22.8 ± 3.8	21.7 ± 4.1	0.029
Severe thinness (n)	0	1	
Thinness (n)	3	2	
Normal (n)	88	95	
Overweight (n)	20	16	
Obesity (n)	7	4	

For continuous variables, differences between Boarding and Day students were compared. Independent sample *t*-test was used to assess *p*-value.

**Table 2 medicina-60-01045-t002:** Dietary diversity score at each 24 h dietary recall day and its mean across three 24 h dietary recall days.

Dietary Diversity Scores	Boarding Students (n = 118)	Day Students (n = 118)	*p*-Value
DDS Weekday 1	4.0 ± 1.4	5.1 ± 1.4	<0.001
DDS Weekday 2	3.7 ± 1.5	4.0 ± 1.4	0.164
DDS Weekend	5.1 ± 1.3	5.9 ± 1.1	<0.001
DDS 3-day mean	4.2 ± 0.8	5.0 ± 0.8	<0.001

Differences between Boarding and Day students were compared. Independent sample *t*-test was used to assess *p*-value. DDS—Dietary Diversity Score.

## Data Availability

Data from the study have all been reported in the manuscript. Original data generated from the research will be made available upon reasonable request in an anonymised form in order to avoid revealing the identity of the participants.

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
