# Peer review of "Low Dietary Diversity and Low Haemoglobin Status in Ghanaian Female Boarding and Day Senior High School Students: A Cross-Sectional Study"

_medicina, 2024, doi:10.3390/medicina60071045_

Round 1
Reviewer 1 Report
Comments and Suggestions for Authors
General comments
All comments are present on the manuscript.
Specific comments
Methodology
- References concerning WHO growth charts and definition of anemia used need to be added.
Results
- Authors need to add the sociodemographic variables studied to Table 1 as these may impact results.
- The sentences below Tables 1 and 2 need to be changed to "Independent sample t-test was used to assess P value.
- I did not find Figure 1 in the manuscript in any section. Therefore numbering of Figures need to be corrected.
Discussion
- Limitations of the study need to be added at the end of the discussion. I think the main limitation of the study is choosing only one school to conduct the study. The other limitation is not including the sociodemographic factors, this can be solved by adding these variables to Table 1.
References
- References 16, 17 and 22; authors need to mention the exact date of access to the website including the year.

There are few linguistic and grammar mistakes
Author Response
Reviewer 1
Thank you for making time to review this piece of work. The authors are grateful for your comments which have led to an improvement in the quality and intent of the manuscript. Please, find below the response to the comments raised for your attention.
Comment:
All comments are present on the manuscript.
Response to reviewer comments:
The authors have revised the manuscript following the comments made in the manuscript
Specific comments
Methodology
Comment: References concerning WHO growth charts and definition of anemia used need to be added.
Response to comment: Thank you for the comment. The authors have provided references in the manuscript to support the WHO growth charts and definition of anaemia. These appear as references [20] and [21] in the manuscript. These can be found in the revised manuscript at Lines 138 and 150.
Results
Comment: Authors need to add the sociodemographic variables studied to Table 1 as these may impact results.
Response to comment: Thank you for the comment. The authors agree that the sociodemographic characteristics of the participants obtained could have had more variables. This has been acknowledged as a “Limitation” of the manuscript. This revised manuscript reads as “In terms of sociodemographic variables, only ethnicity was obtained from the students. Nevertheless, other socio-demographic variables such as household size, occupation of parents, education level of parents etc are not expected to influence the dietary diversity of the boarding students. Hence, these variables were not recorded.” This can be seen in the revised manuscript at Lines 375-378. The Figure showing the Ethnicity can be found at the Supplementary information (Figure S2).
Comment: The sentences below Tables 1 and 2 need to be changed to "Independent sample t-test was used to assess P value.
Response to comment: Thank you for the comment. The authors have revised the statement. The new sentence reads “For continuous variables, differences between Boarding and Day students were compared. Independent sample t-test was used to assess p-value”, For Table 1 and “Differences between Boarding and Day students were compared. Independent sample t-test was used to assess p-value. DDS- Dietary Diversity Score”, for Table 2. These can be found in the revised manuscript at Lines 209-210 and 220-221.
Comment: I did not find Figure 1 in the manuscript in any section. Therefore numbering of Figures need to be corrected.
Response to comment: Thank you so much for pointing this out. The authors have corrected the numbering of the Figures in the revised manuscript starting from Figure 1.
Discussion
Comment: Limitations of the study need to be added at the end of the discussion. I think the main limitation of the study is choosing only one school to conduct the study. The other limitation is not including the sociodemographic factors, this can be solved by adding these variables to Table 1.
Response to comment: Thank you for this wonderful comment. The authors have created a limitation section and highlighted the limitation of the study in line with the suggestions made. This can be seen at the “Limitation” section of the manuscript. The sentence reads as “This study is limited by being conducted in a single metropolitan public high school in Ghana. The findings may not be generalisable to other regions or to private schools, where dietary patterns, socioeconomic factors, and school’s funding might differ. The choice of one Senior High School for this study limits the implications of the study to just the school and may not be a full representation of the broader picture in the Greater Accra region of Ghana.” The other limitation reads as “In terms of sociodemographic variables, only ethnicity was obtained from the students. Nevertheless, other socio-demographic variables such as household size, occupation of parents, education level of parents etc are not expected to influence the dietary diver-sity of the boarding students. Hence, these variables were not recorded”. This can be found in the revised manuscript at Lines 370-387.
References
Comment: References 16, 17 and 22; authors need to mention the exact date of access to the website including the year.
Response to comment: Thank you for the comment. The authors have updated the access date of references 16, 17 and 22 as suggested. Due to the addition of extra new references, the numbering of these references have changed to 19, 21 and 22 in the revised manuscript.

Reviewer 2 Report
Comments and Suggestions for Authors
The study provides valuable insights into the dietary diversity and haemoglobin status among Ghanaian female high school students, highlighting significant differences between boarding and day students. The findings point to the need for enhanced nutritional strategies within the school system. However, the limitations related to the study design, generalizability, and potential biases should be addressed in future research to strengthen the conclusions and implications of such studies.
Comments:
1. The study is conducted in a single metropolitan public high school in Ghana. The findings may not be generalizable to other regions or to private schools, where dietary patterns and socioeconomic factors might differ significantly
2. The cross-sectional nature of the study limits the ability to establish causal relationships between dietary diversity and haemoglobin status. Longitudinal studies would be more effective in understanding the impact of dietary patterns over time
3. Although the sample size calculation is presented, the study does not discuss the potential impact of sampling bias introduced by the convenience sampling method. This could affect the representativeness of the findings.
4. The study relies on dietary diversity scores rather than a detailed quantitative analysis of nutrient intake. This limits the ability to assess the specific nutrient deficiencies that may contribute to anaemia.
5. The classification of anaemia using only haemoglobin concentration might overlook other causes of anaemia such as parasitic infections or genetic factors. A more comprehensive assessment, including serum iron and ferritin levels, would provide a better understanding of the underlying causes of anaemia.
6. While the study mentions the use of partial correlation analysis, there is limited discussion on how other potential confounding variables (e.g., menstrual status, socioeconomic status, presence of infections) were controlled for, which could influence the haemoglobin levels.
7. The reliance on self-reported dietary intake can introduce recall bias, particularly in adolescent populations. The accuracy of the dietary recall data is dependent on the participants' memory and honesty.
Author Response
Reviewer 2
Reviewer comments
Thank you for making time to review this piece of work. The authors are grateful for your comments which have led to an improvement in the quality and intent of the manuscript. Please, find below the response to the comments raised for your attention.
Comments and Suggestions for Authors
The study provides valuable insights into the dietary diversity and haemoglobin status among Ghanaian female high school students, highlighting significant differences between boarding and day students. The findings point to the need for enhanced nutritional strategies within the school system. However, the limitations related to the study design, generalizability, and potential biases should be addressed in future research to strengthen the conclusions and implications of such studies.
Comments:
- The study is conducted in a single metropolitan public high school in Ghana. The findings may not be generalizable to other regions or to private schools, where dietary patterns and socioeconomic factors might differ significantly
Response to comment: We have now addressed this limitation in Lines 370 – 372. The sentence reads as “This study is limited by conducted in a single metropolitan public high school in Ghana. The findings may not be generalisable to other regions or to private schools, where dietary patterns, socioeconomic factors, and school’s funding might differ.”
- The cross-sectional nature of the study limits the ability to establish causal relationships between dietary diversity and haemoglobin status. Longitudinal studies would be more effective in understanding the impact of dietary patterns over time
Response to comment: We have now addressed this limitation in Lines 379 – 382. The sentence reads as “The cross-sectional nature of the study also limits the ability to establish causal relationships between dietary diversity and Hb status, and we have suggested that longitudinal studies would be more effective in understanding the impact of dietary patterns over time.”
- Although the sample size calculation is presented, the study does not discuss the potential impact of sampling bias introduced by the convenience sampling method. This could affect the representativeness of the findings.
Response to comment: We acknowledged that whilst random sampling method is the gold standard for a large-scale epidemiology study to limit bias, a convenience sampling method is also the norm for scientific research. However, we must stress that the researchers have no social relationship to the school, hence the bias of selecting students are minimal. Nevertheless, this limitation is addressed in Lines 382 – 383. This reads as “Whilst the convenience sampling method is a common recruitment method, it is not as robust as random sampling method used in large scale epidemiology study.”
- The study relies on dietary diversity scores rather than a detailed quantitative analysis of nutrient intake. This limits the ability to assess the specific nutrient deficiencies that may contribute to anaemia.
Response to comment: We agree with the reviewer and already addressed this in Lines 313– 316. This reads as “Furthermore, a quantitative dietary analysis may have captured more information compared to using dietary diversity analysis, but we were limited by the lack of reliable food composition database for Ghana at the time of this study [24].”
- The classification of anaemia using only haemoglobin concentration might overlook other causes of anaemia such as parasitic infections or genetic factors. A more comprehensive assessment, including serum iron and ferritin levels, would provide a better understanding of the underlying causes of anaemia.
Response to comment: We have followed the WHO’s threshold for diagnosing anaemia. Nevertheless, we acknowledge by simply looking at Hb concentration may not be adequate. Furthermore, we agree that there may be undiagnosed parasitic infections and genetic factors affecting anaemia. Serum iron and ferritin levels are also useful markers for anaemia. For the information of the reviewer, measurement of Hb concentration is more common in the field as it can be measured using handheld analyser. On the other hand, serum iron and ferritin levels are useful in the clinical setting to better understand the underlying causes of anaemia. In the revised manuscript, this reads as “Although known sickle-cell anaemia is an exclusion criterion for participating in this study, it is possible that other undiagnosed parasitic infections in the students may influence the Hb concentration. Despite iron-deficiency anaemia being the most common form of anaemia globally, being female, menstruation, and parasitic infection were also directly associated with the apparently high anaemia prevalence in amongst adolescents in sub-Saharan Africa. This study design could be improved by concurrent measurement of serum iron and ferritin, which would be an indicator of iron-deficiency anaemia.” These can be seen at Lines 316-318 and 350-353.
- While the study mentions the use of partial correlation analysis, there is limited discussion on how other potential confounding variables (e.g., menstrual status, socioeconomic status, presence of infections) were controlled for, which could influence the haemoglobin levels.
Response to comment: The partial correlation analysis mentioned in Line 259-260 was used to control for residential status (Boarding vs Day) as this is the uniqueness of this study design. However, we did not collect data on menstrual status and the presence of infections.
- The reliance on self-reported dietary intake can introduce recall bias, particularly in adolescent populations. The accuracy of the dietary recall data is dependent on the participants' memory and honesty.
Response to comment: The limitation of dietary recall method has now been included in Lines 383 – 387. This reads as “Finally, there is no perfect method to evaluate dietary intake. In this study, the reliance on self-reported dietary intake can introduce recall bias, particularly in adolescent populations. The accuracy of the dietary recall data is dependent on the participants' memory and honesty.”

Reviewer 3 Report
Comments and Suggestions for Authors
Abstract:
· The introduction of the problem (anaemia and its global prevalence) is well addressed. But the connection between dietary diversity and haemoglobin levels could be emphasized more strongly to highlight the study's relevance and originality.
· The methods are briefly mentioned but lack detail in the abstract. For example, it states that a "semi-structured and three 24-hour dietary recalls" were used without explaining why this specific method was chosen. A brief rationale for choosing methods would enhance the abstract's informativeness.
· The results are presented with statistical outcomes suitable for an abstract. But, the presentation of percentages (22% of boarding students vs. 64% of day students with adequate dietary diversity) could be accompanied by a brief comment on the implications of this disparity.
· The lack of a significant correlation between DDS and haemoglobin concentration is an interesting and appropriately highlighted outcome. Still, the significance of this finding in the context of existing literature could be briefly discussed to improve its relevance.
1. Introduction:
· This section establishes the importance of adequate nutrition during adolescence and its critical role in preventing micronutrient deficiencies, particularly anaemia. The relevance of this study to global and local health contexts, given the high prevalence of anaemia in Ghana compared to global averages, is well articulated. But, the introduction could benefit from a more detailed discussion of the specific nutritional deficiencies prevalent among adolescents in Ghana. Including recent statistics or studies could provide a stronger backdrop for the research.
· The structure of the introduction is generally well-organized, following a logical flow from the importance of nutrition in adolescence to the specifics of anaemia and its prevalence. However, some sentences are overly complex and could be simplified for clarity. For example, the sentence spanning lines 36-40 could be split into two to enhance readability and impact.
· The introduction appropriately cites significant studies and data, like the WHO standards and the Ghana Demographic and Health Survey report. But, it relies heavily on just a few sources and could be strengthened by incorporating a wider range of recent literature to frame the problem more comprehensively. This might include studies that specifically discuss dietary habits and their direct implications on health outcomes among adolescents in similar contexts.
· The introduction emphasizes the importance of investigating hemoglobin levels and dietary variety in a school setting—a unique strategy in the Ghanaian environment. However, a succinct analysis of how this study adds to or differs from other research might improve the story. Highlighting the novel approach or research design, if any, would highlight the manuscript's impact on the field.
2. Materials and Methods:
2.1 Study Design
· The manuscript outlines that a cross-sectional study design was conducted at Tema Senior High School between February and April 2023. This timeframe and setting are appropriate for the study's aims. Still, the authors might consider discussing any seasonal variations in diet that could affect dietary diversity to strengthen the contextual understanding of the data.
2.2 Study Site and Sample Size Calculation
· Selecting Tema Senior High School due to its large adolescent student population is rational. But, the explanation of the sample size calculation using Taro Yamane's formula is briefly mentioned but not fully detailed in its application specifics. It would enhance the clarity and reproducibility of the study if the authors included a more detailed walkthrough of the calculation steps or provided a supplementary file with the calculation details.
2.4 Data Collection
· Using a pre-tested semi-structured questionnaire and the combination of self-administered questionnaires with face-to-face interviews by a trained dietitian are strengths of the study. Still, details on how the questionnaire was pre-tested or validated are missing. Providing this information would help evaluate the reliability of the instruments used for data collection.
2.5 Anthropometric and Biochemical Assessment
· The measurements of body weight and height and the calculation of BMI using standard equipment and procedures are appropriate and well-described. The measurements using a digital hemoglobinometer validated against a gold standard are commendable. It would be beneficial if authors included the specific method used for the validation process to enhance methodological transparency.
2.6 Haemoglobin Measurements
· The description of the validation of the digital haemoglobinometer provides confidence in the reliability of the measures. However, a minor clarification on the procedure for blood collection and the handling of samples before analysis would be useful for replicating the study methodology in similar settings.
2.7 Dietary Diversity Score Determination
· The dietary assessment through three 24-hour recalls is a robust method for capturing usual dietary intake. Given the study population, the Minimum Dietary Diversity for Women (MDD-W) approach is appropriate. Nonetheless, the explanation of how food items were categorized into the ten food groups could be expanded. Also, clarifying whether any adjustments or accommodations were made for local dietary habits could improve understanding of the DDS applicability.
3. Results:
· The description of the statistical significance is well done, but the results could be enhanced by providing effect sizes where applicable. This would give a clearer picture of the magnitude of differences observed, which is particularly useful in educational and nutritional studies.
4. Discussion:
· The authors discuss the absence of a significant difference in hemoglobin levels across varying levels of dietary diversity. Still, they do not delve deeply into potential reasons for this observation. It would be beneficial for the discussion to consider potential biological mechanisms or other confounding factors like genetic traits, chronic infections, or other micronutrient deficiencies that might impact hemoglobin levels. Incorporating such considerations could provide a more thorough analysis of the results.
· The manuscript briefly mentions the implications of observing similar anemia rates in both study groups despite their differing dietary intakes. This aspect of the study could be further enriched by discussing the bioavailability of iron from different food sources and the influence of other dietary components that might enhance or inhibit iron absorption. Such an expanded discussion could offer deeper insights into the nutritional dynamics.
· The authors also reference various other studies within their discussion, yet there is a noticeable absence of critical engagement with these references. It would be advantageous to compare and contrast their findings with studies identifying a positive correlation between dietary diversity and hemoglobin levels. Such comparisons could help hypothesize reasons for the discrepancies between their results and those of other studies.
· Also, the discussion could benefit from including perspectives from studies conducted in contexts beyond Ghana or similar environments. This broader perspective would provide a more global context to the findings, potentially revealing universal trends or unique regional discrepancies.
· The section on public health implications is aptly included. Still, it could be expanded to offer more concrete suggestions for policy changes or interventions that schools could implement in light of the study's findings. Possible recommendations could include enhancements to school meal programs or targeted supplementation programs designed to address the specific nutritional deficiencies identified in the study.
· Given the high prevalence of anemia identified in the study, there is room for a more pronounced call to action directed at local health authorities and school administrations. Emphasizing the urgency and potential strategies for addressing this health issue could mobilize more immediate and effective responses.
Comments on the Quality of English LanguageBased on the excerpt from the document, the English language quality of the text is generally good, with a clear expression of ideas and a structured presentation of research findings. The technical terminology and data presentation are appropriate for an academic article. There are minor issues with grammar and style, but these do not impede understanding significantly.
Author Response
Reviewer 3
Thank you for making time to review this piece of work. The authors are grateful for the constructive comments which have led to an improvement in the quality and intent of the manuscript. Please, find below the authors response to the comments raised for your attention.
Comments and Suggestions for Authors
Abstract:
Comment: The introduction of the problem (anaemia and its global prevalence) is well addressed. But the connection between dietary diversity and haemoglobin levels could be emphasized more strongly to highlight the study's relevance and originality.
Response to comment: Thank you for the comment. The authors have revised the introduction of the abstract to strengthen the connection between dietary diversity and haemoglobin. Due to the restriction on the word limit for the abstract, the authors have included the statement “including haemoglobin (Hb) concentration improvement”, in Line 14.
Comment: The methods are briefly mentioned but lack detail in the abstract. For example, it states that a "semi-structured and three 24-hour dietary recalls" were used without explaining why this specific method was chosen. A brief rationale for choosing methods would enhance the abstract's informativeness.
Response to comment: Thank you for this insightful comment. However, due to the restriction on the word limit (250 words) for the abstract, the authors have stated the rationale for the choice of the three 24-hour dietary recalls in the Method section. The new sentence reads as “The use of the three 24-hour dietary recall was for it to provide a broader picture of the dietary intake of the students as against the use of one 24-hour recall.”
Comment: The results are presented with statistical outcomes suitable for an abstract. But, the presentation of percentages (22% of boarding students vs. 64% of day students with adequate dietary diversity) could be accompanied by a brief comment on the implications of this disparity.
Response to comment: Thank you for the comment. The authors have highlighted the implications of this in the conclusion part of the abstract. This reads as “Low dietary diversity in Boarding students highlighted inadequate nutrition provided by school meals. Strategies to increase meal diversity should be prioritised by stakeholders in Ghana’s educational sector.” This can be seen at Lines 27-29.
Comment: The lack of a significant correlation between DDS and haemoglobin concentration is an interesting and appropriately highlighted outcome. Still, the significance of this finding in the context of existing literature could be briefly discussed to improve its relevance.
Response to comment: Thank you for this comment. Due to the restriction on the word limit for the abstract, the authors have highlighted the significance of this finding in the context of existing literature in the Discussion section.
- Introduction:
Comment: This section establishes the importance of adequate nutrition during adolescence and its critical role in preventing micronutrient deficiencies, particularly anaemia. The relevance of this study to global and local health contexts, given the high prevalence of anaemia in Ghana compared to global averages, is well articulated. But, the introduction could benefit from a more detailed discussion of the specific nutritional deficiencies prevalent among adolescents in Ghana. Including recent statistics or studies could provide a stronger backdrop for the research.
Response to comment: The authors have revised the introduction by highlighting other micronutrient deficiencies in non-pregnant adolescent women in Ghana from a National survey that was carried out in 2017. This is to complement the findings from the recent report by the Ghana Demographic and Health Survey report. The new sentence reads as “In the year 2017, a national survey which took the form of a cross-sectional study was conducted using non-pregnant adolescent women in Ghana aged 15-49 years to determine micronutrient deficiencies [9]. The authors reported that the prevalence of anaemia, vitamin A deficiency, folate deficiency and vitamin B12 deficiency stood at 21.7%, 1.5%, 53.8%, and 6.9% respectively [9].” Also, the authors have highlighted this sentence “Based on the same threshold, the recent Ghana Demographic and Health Survey re-port revealed that 41% of non-pregnant Ghanaian women aged 15-49 years were anaemic [11], higher than the global average. Another recent cross-sectional study in Ghana involving an analysis of data from students comprising 2948 adolescent females and 609 males aged 10–19 years showed anaemia prevalence of 24 and 13% respectively [12].” These can be found in the revised manuscript at Lines 44-48 and 53-57.
Comment: The structure of the introduction is generally well-organized, following a logical flow from the importance of nutrition in adolescence to the specifics of anaemia and its prevalence. However, some sentences are overly complex and could be simplified for clarity. For example, the sentence spanning lines 36-40 could be split into two to enhance readability and impact.
Response to comment: Thank you for the comment. The authors have split the sentence spanning lines 37-48 could be split into two to enhance readability and impact as suggested. The new sentence with the split paragraph reads as:
“Children and adolescents who commonly consume insufficient animal-based foods, fruits and vegetables, but excessive convenient, energy-dense depleted foods [3] may suffer from hidden hunger characterised by micronutrient deficiency [4,5]. Occasionally, they may be apparently “well-fed”, hence overweight, but also suffer from micronutrient deficiency [6]. Consequently, a high-quality diet is characterised by consuming foods form diverse food groups to ensure sufficient intake of various micronutrients [7,8].”
In the year 2017, a national survey which took the form of a cross-sectional study was conducted using non-pregnant adolescent women in Ghana aged 15-49 years to determine micronutrient deficiencies [9]. The authors reported that the prevalence of anaemia, vitamin A deficiency, folate deficiency and vitamin B12 deficiency stood at 21.7%, 1.5%, 53.8%, and 6.9% respectively [9].”
Comment: The introduction appropriately cites significant studies and data, like the WHO standards and the Ghana Demographic and Health Survey report. But, it relies heavily on just a few sources and could be strengthened by incorporating a wider range of recent literature to frame the problem more comprehensively. This might include studies that specifically discuss dietary habits and their direct implications on health outcomes among adolescents in similar contexts.
Response to comment: The authors have added this sentence to the latter part of the introduction to strengthen it “In China, it has been reported that dietary staples for Boarding students are pre-dominantly rich in carbohydrates and inferior in protein-rich food. This may potentially predispose the students to anaemia as proteins remain an integral component of haematinic diets especially when taken with vitamin C [17]. In Ghana, it has been established that poor dietary diversity score is not an important predictor of anaemia among children aged 6–59 months in Ghana [18]. What is however unknown is the association between dietary diversity score and anaemia in adolescents Boarding and Day students.” This can be found in the revised manuscript from Lines 78-84.
Comment: The introduction emphasizes the importance of investigating hemoglobin levels and dietary variety in a school setting—a unique strategy in the Ghanaian environment. However, a succinct analysis of how this study adds to or differs from other research might improve the story. Highlighting the novel approach or research design, if any, would highlight the manuscript's impact on the field.
Response to comment: Thank you for the comments. The authors have revised the last paragraph of Introduction (Line 87 – 92) to address the novel approach of the study, “Whilst there are existing data on the association between dietary diversity and health status, this study uniquely introduces the residential status of students as a contributing factor to dietary diversity and health status. Despite the students attended the same education environment, the food environment between Boarding and Day students may be different. This study will offer novel understanding on the importance of school’s meal provision on dietary diversity of the Boarding students.”
- Materials and Methods:
2.1 Study Design
Comment: The manuscript outlines that a cross-sectional study design was conducted at Tema Senior High School between February and April 2023. This timeframe and setting are appropriate for the study's aims. Still, the authors might consider discussing any seasonal variations in diet that could affect dietary diversity to strengthen the contextual understanding of the data.
Response to comments: Thank you for this comment. The authors have added “Also, seasonal variations in diet may affect the dietary diversity of the students”, to the discussion section of the manuscript.” Lines 300-301.
2.2 Study Site and Sample Size Calculation
Comment: Selecting Tema Senior High School due to its large adolescent student population is rational. But, the explanation of the sample size calculation using Taro Yamane's formula is briefly mentioned but not fully detailed in its application specifics. It would enhance the clarity and reproducibility of the study if the authors included a more detailed walkthrough of the calculation steps or provided a supplementary file with the calculation details.
Response to comment: Thank you for the comment. The authors have expanded the abbreviations used in the formula (Line 109 – 114) to provide clarity to readers.
2.4 Data Collection
Comment: Using a pre-tested semi-structured questionnaire and the combination of self-administered questionnaires with face-to-face interviews by a trained dietitian are strengths of the study. Still, details on how the questionnaire was pre-tested or validated are missing. Providing this information would help evaluate the reliability of the instruments used for data collection.
Response to comment:
Thank you for the comments. The semi-structured questionnaire was used to collect information on student’s residential status, ethnicity, age, anthropometric parameters, biochemical parameters and 24-hour dietary recall and hence no validation was required. “Pre-test” was supposed to mean that researchers were trained to collect data using this semi-structured questionnaire prior to collect real data from the participants. The authors apologise for the confusion as “pre-tested” here did not add any useful information to the methods, hence we decided to avoid using the word “pre-tested”.
2.5 Anthropometric and Biochemical Assessment
Comment: The measurements of body weight and height and the calculation of BMI using standard equipment and procedures are appropriate and well-described. The measurements using a digital hemoglobinometer validated against a gold standard are commendable. It would be beneficial if authors included the specific method used for the validation process to enhance methodological transparency.
Response to comment: Thank you for the comment. The authors have attached the correlation analysis plot as requested as a Supplementary material (Figure S1).
2.6 Haemoglobin Measurements
Comment: The description of the validation of the digital haemoglobinometer provides confidence in the reliability of the measures. However, a minor clarification on the procedure for blood collection and the handling of samples before analysis would be useful for replicating the study methodology in similar settings.
Response to comment: Thank you for the comment. The authors have provided some deeper information regarding how the haemoglobinometer was used for the Hb concentration analysis as suggested. The new sentence reads as “In the present study, venous blood samples of students were collected by a phlebotomist at the school premises into EDTA vacutainer. The Hb concentrations of the anti-coagulated blood were determined using the validated digital haemoglobinometer in measuring time of Ë‚ 10 sec.” This can be found in the revised manuscript at Lines 146-149.
2.7 Dietary Diversity Score Determination
Comment: The dietary assessment through three 24-hour recalls is a robust method for capturing usual dietary intake. Given the study population, the Minimum Dietary Diversity for Women (MDD-W) approach is appropriate. Nonetheless, the explanation of how food items were categorized into the ten food groups could be expanded. Also, clarifying whether any adjustments or accommodations were made for local dietary habits could improve understanding of the DDS applicability.
Response to comment: Thank you for the comments. The authors have uploaded a Table with the categorized food items at the Supplementary information section (Table S1) for your attention as requested.
- Results:
Comment: The description of the statistical significance is well done, but the results could be enhanced by providing effect sizes where applicable. This would give a clearer picture of the magnitude of differences observed, which is particularly useful in educational and nutritional studies.
Response to comment: Thank you for the comment. The authors have revised the manuscript highlighting that “effect size was presented as odds ratio (OR) with 95% confidence interval (CI)” at the statistical analysis section. The result section has been revised to reads as “The Boarding students were less likely to consume nuts and seeds (OR: 0.36, 95%CI [0.20 – 0.66], p < 0.001), dairy (OR: 0.45, 95%CI [0.29 – 0.70], p < 0.001), flesh foods (OR: 0.38, 95%CI [0.28 – 0.50], p < 0.001), eggs (OR: 0.59, 95%CI [0.41 – 0.85], p = 0.005), vitamin A-rich vegetables and fruits (OR: 0.49, 95%CI [0.35 – 0.69], p < 0.001), other vegetables (OR: 0.57, 95%CI [0.38 – 0.85], p = 0.006), and other fruits (OR: 0.21, 95%CI [0.10, 0.43], p< 0.001) compared to the Day students”. This can be seen in the revised manuscript at Lines 191-192 and 227-231.
- Discussion:
Comment: The authors discuss the absence of a significant difference in hemoglobin levels across varying levels of dietary diversity. Still, they do not delve deeply into potential reasons for this observation. It would be beneficial for the discussion to consider potential biological mechanisms or other confounding factors like genetic traits, chronic infections, or other micronutrient deficiencies that might impact hemoglobin levels. Incorporating such considerations could provide a more thorough analysis of the results.
Response to Comments:
Thank you for the comments. We have actually addressed the potential reasons for the lack of significant correlation between Hb and dietary diversity in Lines 310 – 316, “This could be explained that low Hb concentration is usually developed over a prolonged period of undernutrition hence may not be simply captured by dietary recalls at a single point in time. Long-term dietary diversity may be a better indicator of Hb concentration. Furthermore, a quantitative dietary analysis may have captured more information compared to using dietary diversity analysis, but we were limited by the lack of reliable food composition database for Ghana at the time of this study.”
We appreciate the reviewer’s insight into the role of infections, hence we have now also added that “Although known sickle-cell anaemia is an exclusion criterion for participating in this study, it is possible that undiagnosed parasitic infections in the students may influence the Hb concentration.”
Comment: The manuscript briefly mentions the implications of observing similar anemia rates in both study groups despite their differing dietary intakes. This aspect of the study could be further enriched by discussing the bioavailability of iron from different food sources and the influence of other dietary components that might enhance or inhibit iron absorption. Such an expanded discussion could offer deeper insights into the nutritional dynamics.
Response to comment: Thank you for the comments. The authors have revised the manuscript to reflect potential causes of the high anaemia prevalence amongst the adolescent students. The new sentence read as “In Ghana, muscle meat which contains bioavailable form of haem iron is relatively expensive. In contrast, plant-based alternative sources, especially from indigenous green leafy vegetables tend to be the common source of non-haem iron for the older population. However, in recent times, there has been a general decline in indigenous leafy vegetable intake particularly amongst adolescent Ghanaians. This could have contributed to the high anaemia prevalence. In addition to the low intake of the leafy vegetables, factors including poor cooking practices for indigenous vegetables, over-cooking of indigenous vegetables and poor intake of vitamin C-rich fruit decreases the bioavailability of the iron for absorption.” This can be found at Lines 336-345.
Comment: The authors also reference various other studies within their discussion, yet there is a noticeable absence of critical engagement with these references. It would be advantageous to compare and contrast their findings with studies identifying a positive correlation between dietary diversity and hemoglobin levels. Such comparisons could help hypothesize reasons for the discrepancies between their results and those of other studies.
Response to comment: Thank you for the comment. We have expanded the discussion based on the references. We highlighted that in all these studies (including ours), <50% of participants had adequate dietary diversity, but >50% of participants had anaemia based on Hb concentration. Although these do not help us to understand why there is a lack of positive association between dietary diversity and Hb concentration, we made a stronger implication by addressing that low dietary diversity may put them at risk of future anaemia. Therefore, a longitudinal observation may have been more informative. The authors have revised Lines 328-332 to address this.
Comment: Also, the discussion could benefit from including perspectives from studies conducted in contexts beyond Ghana or similar environments. This broader perspective would provide a more global context to the findings, potentially revealing universal trends or unique regional discrepancies.
Response to comment: Thank you for the comment. The authors have compared the findings with that reported for a study in China. The authors have highlighted this in yellow “This observation is consistent with a previous report in China that school meals for Boarding students may be monotonous and rich in carbohydrate staples [17]. Therefore, it becomes a voluntary choice for Boarding students to purchase additional foods to improve their dietary diversity. Nevertheless, the consumption of nuts and seeds, dairy, eggs, dark green leafy vegetables, and other fruits were generally low in this co-hort, which is also consistent with that reported other sub-Saharan Africa school-going adolescents [22].” This can be found in the revised manuscript at Lines 287-293.
Comment: The section on public health implications is aptly included. Still, it could be expanded to offer more concrete suggestions for policy changes or interventions that schools could implement in light of the study's findings. Possible recommendations could include enhancements to school meal programs or targeted supplementation programs designed to address the specific nutritional deficiencies identified in the study.
Response to comment: Thank you for the comment. The new sentence reads as “There is a need to enhance the nutritional profile of meals served to senior high school students by incorporating foods rich in bioavailable iron. Additionally, school food producers should be trained on strategies for cooking plant-based legumes, which can enhance the digestibility of proteins. Promoting the increased consumption of dark green indigenous vegetables should also be a priority, as these vegetables provide relatively cheaper sources of iron. In recent times, fish farming especially involving catfish and tilapia rearing is on the rise in Ghana. Schools can be encouraged to adopt that as part of their science curriculum which could provide a pathway for fish meal powders which could be promoted for its use in cooking recipes. Lastly, adding vitamin C-rich fruits to school-based meals should be encouraged to enhance the absorption of dietary iron.” This can be found in the revised manuscript at Lines 409-418.
Comment: Given the high prevalence of anemia identified in the study, there is room for a more pronounced call to action directed at local health authorities and school administrations. Emphasizing the urgency and potential strategies for addressing this health issue could mobilize more immediate and effective responses.
Response to comment: Thank you for the comment. The authors have included a “Recommendation” section that has pragmatically highlighted strategies that can address the apparent high prevalence of anaemia. The sentence reads as “There is a need to enhance the nutritional profile of meals served to senior high school students by incorporating foods rich in bioavailable iron. Additionally, school food producers should be trained on strategies for cooking plant-based legumes, which can enhance the digestibility of proteins. Promoting the increased consumption of dark green indigenous vegetables should also be a priority, as these vegetables provide relatively cheaper sources of iron. In recent times, fish farming especially involving catfish and tilapia rearing is on the rise in Ghana. Schools can be encouraged to adopt that as part of their science curriculum which could provide a pathway for fish meal powders which could be promoted for its use in cooking recipes. Lastly, adding vitamin C-rich fruits to school-based meals should be encouraged to enhance the absorption of dietary iron.” This can be found in the manuscript at Lines 409-418.
Comments on the Quality of English Language
Comment: Based on the excerpt from the document, the English language quality of the text is generally good, with a clear expression of ideas and a structured presentation of research findings. The technical terminology and data presentation are appropriate for an academic article. There are minor issues with grammar and style, but these do not impede understanding significantly.
Response to comment: Thank you for the comment. The authors have thoroughly revised the manuscript to address grammatical challenges.
